# MYCN-Driven Metabolic Networks Are a Critical Dependency of High-Risk Neuroblastomas

**DOI:** 10.3390/cancers17193256

**Published:** 2025-10-08

**Authors:** Michelle G. Pitts, Lindsay T. Bryant, Michael D. Buoncristiani, Eric J. Rellinger

**Affiliations:** 1Department of Surgery, University of Kentucky, Lexington, KY 40536, USA; michelle.pitts@uky.edu (M.G.P.); lindsay.bryant1231@uky.edu (L.T.B.); michael.buoncristiani@uky.edu (M.D.B.); 2Markey Cancer Center, University of Kentucky, Lexington, KY 40536, USA

**Keywords:** neuroblastoma, metabolism, MYCN, glycosylation, glycolysis, fatty acids, polyamines, sialic acid, fucosylation

## Abstract

**Simple Summary:**

Neuroblastoma is a deadly childhood cancer that is more aggressive when the *MYCN* gene is turned on. Directly targeting *MYCN* has been very difficult. Overactive *MYCN* changes the way that cancer cells use nutrients, and teams of scientists have been working to understand how these changes may be used to fight neuroblastomas. This review outlines a number of the metabolic changes that are affected by *MYCN* amplification, including glycolysis, amino acid dependencies, polyamines, and aberrant glycosylation, and attempts to provide clinically relevant insight into their interconnectedness. Moreover, we highlight glycosylation as an emerging area of study in the context of neuroblastoma and summarize current efforts towards understanding how changes in carbohydrate display may affect neuroblastoma presentation and treatment.

**Abstract:**

Neuroblastoma is a devastating pediatric solid tumor that, despite significant recent advances, still accounts for nearly 15% of all childhood cancer deaths. Patients are risk stratified based on a number of features, including amplification of the *MYCN* oncogene, yet targeting *MYCN* itself has been unsuccessful to date. The complex interplay between this oncogene and its many metabolic targets has proven challenging and is only beginning to be understood in the context of pediatric tumors. It is increasingly recognized, however, that *MYCN*-driven metabolic rewiring and concomitant increases in biosynthetic precursors has the potential to drive many aspects of tumor development. Furthermore, emerging research suggests that improving overall therapeutic outcomes for neuroblastoma patients may well require individual metabolic profiling, allowing personalized simultaneous targeting of multiple metabolic nodes. In this review, we outline clinically relevant research involving *MYCN*-driven metabolic derangements, including increased glucose uptake, polyamine synthesis, glycosylation, and others, and attempt to summarize the influence of *MYCN* on important metabolic genes and druggable protein targets. We spotlight emerging research in glycosylation and its modulation as an often overlooked but increasingly promising therapeutic area. It is our hope that this document will provide utility for both clinicians and scientists seeking to understand how the *MYCN* oncogene and metabolism are critically intertwined.

## 1. Introduction

Neuroblastomas (NBs) arise from the sympathetic chain and are the most common extracranial solid tumor in children (~700 cases per year) [1]. Neuroblastoma is a histologically and clinically heterogeneous disease with disease progression ranging from spontaneous metastatic regression to overwhelming disease progression despite aggressive treatment with induction chemotherapy, surgery, radiation therapy, and immunologic therapy [2]. Children are risk stratified based upon age, extent of spread, histologic features, chromosomal aberrations, and *MYCN*-amplification [1,2]. *MYCN*-amplified (*MYCN*-amp) NBs account for 40% of children with high-risk disease characterized by extra copies of the *MYCN* gene detected by fluorescent in situ hybridization [3]. *MYCN* is a member of the MYC oncogenic transcription factor family (along with C-MYC and L-MYC) known to support cancer development and progression by driving proliferation, blocking differentiation, promoting genomic instability, inducing angiogenesis, and facilitating immune evasion [4,5]. *MYCN*-driven metabolic rewiring contributes to these cancer hallmarks by increasing biosynthetic precursors necessary to sustain proliferation and replicative immortality, altering redox balance to both resist cell death and drive genomic instability, and altering the tumor microenvironment to facilitate immune evasion [5].

Owing to its ubiquitous oncogenic functions across a wide array of cancers, C-MYC is a more common focus of cancer metabolism studies [4,5,6,7]. N-MYC is less ubiquitously expressed and commonly constrained to expression within tissues of neuroectodermal-derived many cancers, including NB, medulloblastoma, retinoblastoma, and small cell lung cancers [8]. C-MYC and N-MYC have numerous overlapping roles in their metabolic functions both driving glycolysis, amino acid dependence, context-dependent oxidative phosphorylation, ribosomal biogenesis, and fatty acid uptake (Figure 1) [4,5]. Metabolic intermediates (such as acetyl-CoA, alpha ketoglutarate, NAD+, etc.) can have drastic implications for how epigenetic machinery affects cellular states and plasticity, which is increasingly recognized to facilitate disease progression and therapeutic resistance [9,10]. Similarly, MYC-driven metabolic shifts have been shown to alter how post-translational modifications, including glycosylation, may be highjacked in cancer development and progression [11,12,13]. Glycosylation aberrancies contribute significantly to the heterogeneity of proteins and lipids, altering how these macromolecules are distributed, degraded, activated, and ultimately contribute to the interactome of cancer cells with their local tumor microenvironment [14,15]. These common threads demonstrate how oncogenic MYC signaling can contribute to sweeping metabolic changes that contribute to cancer cell progression and therapeutic failure.

Herein, we seek to provide a central, but not all-encompassing, resource for both clinicians and scientists to understand how *MYCN*-driven metabolic perturbations have been shown to contribute to NB tumor formation, progression, and treatment failure. We attempt to summarize the key *MYCN*-driven perturbations within Table 1. The majority of these studies are limited to in vitro conditions where cancer cells are frequently cultured under artificial conditions that may not represent the true biology observed within our patients. Increasing utilization of advanced metabolomic techniques, including stable-isotope tracing to resolve metabolite flux and spatial metabolomics, in humans and human tissues samples, is likely to be a fruitful route toward developing new treatment strategies for this metabolically-active and hard-to-treat childhood cancer [16,17].

## 2. Oxidative Phosphorylation and Glycolysis

While oxidative phosphorylation produces ATP much more efficiently, cancer cells often rewire their metabolism towards glycolysis, even in the presence of oxygen, a phenomenon known as the Warburg effect [35,36]. This metabolic shift promotes increased glucose uptake, which ultimately supports macromolecule synthesis necessary for tumor progression. The Warburg effect allows the formation of key glycolytic intermediates, including glucose 6-phosphate, 3-phosphoglycerate, and pyruvate, supporting downstream N-linked glycosylation, lipid formation, and the production of glutathione, among others [5,36,37]. The central role of preferential glycolysis in many cancers is widely recognized; however, the regulatory landscape and potential therapeutic interventions remain poorly understood and under-investigated, particularly in pediatric cancers, underscoring a critical need for further investigation [37].

Thirty years ago, PET imaging demonstrated that NBs display increased glucose uptake relative to normal tissues [38]. More recently, intraoperative ^13^C glucose tracing in a small cohort of tumors revealed elevated lactate labeling in NBs compared to other tumor types, suggesting increased glycolytic activity in NBs relative to other pediatric tumors. Notably, only one of the NBs in this cohort was *MYCN*-amp [39]. Myc activation is known to increase glucose uptake, drive mitochondrial biogenesis, and initiate metabolic programs that use both oxidative phosphorylation and glycolysis [6,7,40]. In vitro work using *MYCN* ON/OFF NB cells has also demonstrated that *MYCN* activation in cells under stress can shift ATP production to mitochondrial oxidation of fatty acids, while aerobic glycolysis was also increased under stress but independent of *MYCN* status. Accordingly, limiting fatty acid oxidation decreased NB cell viability and tumor burden in mice [41]. Overall, however, the role of oxidative phosphorylation in *MYCN*-amp NB and its specific targeting remains poorly understood.

Hexokinase 2 (HK2) is the enzyme that catalyzes the first committed step of glycolysis by phosphorylating glucose to yield glucose-6-phosphate. HK2 is upregulated by *MYCN* induction in NB cells, along with a shift toward use of glycolysis and concomitant sensitization to 2-deoxyglucose [18]. *MYCN*-amp cells and primary tumors have also been demonstrated to express hypoxia inducible factor 1α (HIF1α), which cooperates with N-Myc to regulate both *HK2* and *LDHA* [20]. HK2 is tumor-promoting in many cancers and is significantly upregulated in *MYCN*-amp primary tumors, making it an attractive target for inhibition [20,37]. 3-bromopyruvate, an inhibitor of HK2, has demonstrated synergism with rapamycin, an mTOR inhibitor in vitro [19]. Interestingly, “escape” of glycolytic dependency depends on mTORC1 expression, and some PI3K/mTOR inhibitors were also found to destabilize *MYCN* in vitro and in mouse models [42,43]. Together, these data demonstrate that a sustained effect when targeting glycolysis may only be accomplished by considering the many metabolic and survival nodes that these enzymes and intermediates can affect.

Increased expression of lactate dehydrogenase A (LDHA), which catalyzes the conversion of pyruvate to lactate, is associated with poor patient outcomes [44,45]. Results are mixed though regarding its overall role in MYCN-amp tumors. LDHA was shown to be required for normoxic *MYCN*-amp NB proliferation and tumorigenesis [20]; however, a study using patient data, CRISPR/Cas9 *LDHA* knockdown, and TH-*MYCN* mice indicated that LDHA expression was independent of *MYCN* and was actually dispensable for aerobic glycolysis, suggesting important alternative roles for the enzyme [44]. Several small-molecule LDHA inhibitors have been described, including FX11, which disrupts aerobic glycolysis, reduces cell viability and proliferation, and induces apoptosis in *MYCN*-amp NB cell lines, and Oxamate, a competitive inhibitor that can be delivered to tumors via targeted liposomes. However, preclinical studies remain limited, and the precise relationship among MYCN, LDHA, and the Warburg effect in NB is still unclear [21,45].

Recent work has emerged indicating that the combination of a mitochondrial uncoupler along with retinoic acid, a common NB treatment, would reverse the Warburg effect and promote more efficient NB differentiation than retinoic acid alone by activating mitochondrial respiration [46]. While overall successful, in vivo application produced only partial differentiation, likely owing to tumor-wide heterogeneity [46]. Indeed, work in lung cancer models has highlighted metabolic heterogeneity and subtype-specific metabolic alterations, suggesting that patients with many tumor types may benefit from metabolic studies to better understand the state of their tumor as well as its response to treatment [47,48].

## 3. MYCN-Regulated Amino Acid Dependencies

Deregulated *MYC* favors not only increased glucose uptake and energy and macromolecule generation, but also an increased dependency on certain amino acids to use as building blocks as well as replenish key intermediates in the tricarboxylic acid cycle [5,49]. Glutamine, serine, and cysteine act as central metabolic mediators, supplying carbon and nitrogen for nucleotide, protein, and lipid synthesis, and have been among the most extensively studied amino acids linked to *MYCN*-driven NB.

Glutamine is a conditionally essential amino acid that plays diverse roles in cell metabolism and growth, spanning from energy generation to end-product macromolecule synthesis [49,50]. N-Myc activates expression of the glutamine transporter SLC1A5/ASCT2 as well as mitochondrial glutaminase 2 (GLS2), which catalyzes the formation of glutamate from glutamine, is elevated in *MYCN*-amp tumors, and correlates with poor survival [22,23]. Glutamate can be converted to α-ketoglutarate and further used in the citric acid cycle or used for biosynthesis of glutathione, an important antioxidant. Glutathione biosynthesis was demonstrated in TH-*MYCN*^++^ mice as upregulated to promote cell survival during very early tumorigenesis [51]. This study further suggested that coupling existing chemotherapies with glutathione biosynthesis inhibition may potentiate the therapy. GLS2 and SLC1A5 inhibition or glutamine deprivation have been shown to trigger significant inhibition of aerobic glycolysis and cell death [22,23,52]. However, *MYCN*-amp NB cells have also been shown to synthesize glutamine de novo and survive glutamine starvation in the presence of glucose, hinting at the extreme metabolic adaptability of these cells [41]. High SLC1A5 expression has also recently reported to be correlated with immune cell infiltration and demonstrated that the SLC1A5 inhibitor V-9032 may also regulated ST8SIA1 expression within NB cells [53]. This study invites further inquiry into determining how glutamine uptake may be implicated into NB immune evasion within preclinical models.

N-Myc activates ATF4 mRNA, thus stabilizing N-Myc and allowing formation of a positive feedback loop necessary for transcriptional activation of the serine–glycine–one-carbon (SGOC) pathway [24]. This is a highly interconnected pathway, and for a review that includes its roles in cancer, we recommend reading Reina-Campos et al. [54]. At the heart of SGOC, glucose is used to generate serine and glycine. These two then contribute one-carbon units to the folate cycle, feeding into multiple other points that sustain homeostasis. Serine can also contribute to purine biosynthesis. Amplification of *MYCN* creates a SGOC gene signature consisting of *PHGDH*, *PSAT1*, *MTHFD1L*, *MTHFD1*, *MTHFD2*, and *SHMT2***,** all of which are elevated in *MYCN*-amp cases compared to either low-risk or high-risk tumors without amplification [24]. Pharmacological inhibition of phosphoglycerate dehydrogenase (PHGDH), one of the enzymes necessary for the conversion of glucose to serine, was shown to trigger metabolic stress and G1 arrest specifically in *MYCN*-amp NB cells while non-amp cells were unaffected [24]. In vitro and in vivo trials demonstrate increased susceptibility of MYCN-amplified NBs to NCT-503, a small molecule inhibitor of PHGDH, suggesting that MYCN-amplified NBs are preferentially sensitive to the SGOC therapeutic strategies [24].

*MYCN*-amp cells also have an elevated requirement for folate, and the combination of *MYCN* inhibition and either silencing of methylenetetrahydrafolate dehydrogenase 1 (MTHFD1) or methotrexate treatment of *MYCN*-amp cells alters redox homeostasis, causes apoptosis in vitro, and significantly slows xenograft growth [25,26]. Knockdown or pharmacological inhibition of two *MYCN*-controlled enzymes involved in purine biosynthesis also demonstrated robust effects on *MYCN*-amp cell viability in vitro, suggesting that targeting SGOC may be a potent strategy that limits availability of multiple metabolites [55].

Given the fact that targeting *MYCN* alone has proven unsatisfying, the crosstalk of *MYCN*-driven amino acid dependencies with other biological processes may prove extremely useful. For example, depletion of cysteine, necessary for glutathione synthesis, in *MYCN*-amp cells was shown to trigger ferroptosis resulting from accumulated reactive oxygen species [56]. A feed-forward loop also exists between *MYCN* and the amino acid transporters SLC7A5 and SLC43A1, and specific inhibition of these transporters inhibits both glutamine and glucose metabolism while also inhibiting N-Myc synthesis [57]. Overall, however, very few therapies targeting amino acid dependencies have been successfully developed, and to the best of our knowledge, none have been clinically implemented in NB.

A major challenge to the translational integration of targeting amino acid dependencies is that much of this foundational work derives from studies in two-dimensional cultured cell systems, where nutrient availability, oxygen tension, and immune interactions differ dramatically from the tumor microenvironment in vivo and in situ. For instance, the composition of amino acids and redox buffers in standard culture media (e.g., high cystine, glutamine, and glucose) does not accurately reflect the nutrient gradients, stromal interactions, or hypoxic conditions present within bulky NBs [58]. As a result, metabolic dependencies observed in vitro may be exaggerated, or alternatively masked, compared to what occurs in patients, highlighting the need to develop physiologically relevant NB culture systems and highlighting the importance of validating cell culture within disease relevant models in vivo.

## 4. Polyamine Synthesis and Uptake

Polyamines are small organic compounds with at least two amino groups. They are essential for numerous processes in cell growth and differentiation and, as such, their synthesis is typically upregulated in fast-growing tumors such as NB [27,59]. Multiple enzymes involved in polyamine synthesis, including rate-limiting ornithine decarboxylase (*ODC1*), are dysregulated in *MYCN*-amp NB and, furthermore, high-risk tumors without *MYCN* amplification also feature high levels of ODC1, suggesting that targeting this enzyme may provide broad benefit in NB treatment [60,61].

*MYCN* amplification promotes increased polyamine synthesis by directly targeting *ODC1* transcription as well as by repressing antizyme breakdown of ODC and polyamine catabolism [27,44]. Moreover, *ODC1* has been shown in a subset of tumors to be co-amplified along with *MYCN* [27]. Tumor cells can also take up polyamines from the microenvironment via SLC3A2, a direct N-Myc target, and can compensate for single-agent inhibition of polyamine synthesis by upregulating a second transporter, ATP13A3 [28,62]. Both transporters have been shown to be druggable with the small molecule polyamine transport inhibitor AMXT 1501 [62,63].

Difluoromethylornithine (DFMO) was approved by the US FDA in late 2023 to target polyamine synthesis via ODC1 inhibition in high-risk post-immunotherapy patients in remission (NCT02395666). This maintenance therapy has been shown to increase both event-free and overall survival in high-risk *MYCN*-amp and non-amp cases [64,65]. DFMO is undergoing numerous clinical trials for additional indications and drug combinations, including AMX 1501, standard of care chemotherapies, and dinutuximab. Combining DFMO with celecoxib, which upregulates spermine/spermidine acetyltransferase (*SAT1*), thus regulating polyamine cellular export and depleting polyamines, is also under investigation [28,61,66]. DFMO treatment with restricted dietary arginine intake may also increase the benefit of DFMO treatment by limiting ornithine generation and thus tumor uptake, highlighting the role that careful examination of exogenous sources of important intermediates can play in cancer treatment [67]. DFMO also represses Lin28, an RNA binding protein that promotes stemness, by enhancing synthesis of let7, a tumor suppressor microRNA that is repressed by N-Myc [68]. This inhibitory axis regulates glycolytic metabolism, discussed earlier in this review, thus suggesting that DFMO treatment could have patient benefits on multiple fronts [69].

## 5. Fatty Acid Dependencies

Tumors reactivate de novo fatty acid synthesis in order to support rapid cell division and overall growth, which requires increased availability of lipids to form membranes, perform signaling functions, and serve as energy sources [70]. Multiple studies have shown a concerted role for c-Myc in fatty acid metabolism and abundance, but a role for *MYCN* and its amplification is generally less clear [29].

Fatty acids serve as the basis for more complex lipids. De novo fatty acid synthesis requires acetyl-CoA carboxylase and fatty acid synthase (FASN), a direct target of N-Myc, to produce palmitate, which is then modified further to produce other fatty acids [30,70]. *MYCN* additionally promotes de novo fatty acid synthesis through glucose-sensing transcription factor MondoA and lipogenic transcription factor SREBP1 [71,72]. MondoA loss significantly abrogates glutamine-derived lipid synthesis, and N-Myc overexpressing cells are particularly sensitive to in vitro FASN inhibition when MondoA is knocked down [71]. FASN inhibition was shown to reduce NB xenograft growth and increase differentiation through activation of ERK signaling with concomitant reductions in both c-Myc and N-Myc; however, those benefits were found to be independent of *MYCN* amplification [29].

Fatty acids can also be taken up by cells from their environment through specific transporters [70]. Moreover, NB cells use exogenous fatty acids to evade FASN inhibition [31]. *MYCN* promotes fatty acids uptake by NB cells through upregulation of the fatty acid transporter FATP2, encoded by *SLC7A2*, and pharmacological inhibition of this uptake in an animal model significantly impacts tumor growth and survival by targeting FATP2 overexpressing cells [31]. Interestingly, FATP2 has also been implicated in tumor neutrophil-derived myeloid suppressor cell activity, indicating that targeting lipid mediator formation may produce significant results in the tumor microenvironment [73].

In vitro work has demonstrated *MYCN*-amp NB cells to be highly dependent on fatty acid β-oxidation for growth and survival; however, rigorous validation in pre-clinical mouse models remains limited [41,74]. Several points of the β-oxidation pathway have emerged as promising therapeutic vulnerabilities in other cancers, suggesting that targeting lipid metabolism may hold untapped potential in NB as well [75]. At the same time, inhibition of fatty acid synthesis represents a particularly compelling strategy within the context of *MYCN*-driven metabolic reprogramming, as it may simultaneously restrict energy production and membrane biosynthesis. Despite this promise, the current pharmacological toolkit remains underdeveloped. Expanding this toolkit will require both adaptation of existing metabolic inhibitors from other cancer settings and development of NB-tailored approaches guided by in vivo validation. However, major challenges remain, including limited specificity, systemic toxicity given the importance of lipid metabolism in normal tissues, and the intrinsic redundancy of metabolic networks that allows NB cells to adaptively rewire their fuel utilization and bypass single-pathway inhibition [76,77]. Overcoming these barriers will necessitate rational combination strategies, improved drug design, and integration of metabolic biomarkers into translational studies.

## 6. Glycosylation

Glycosylation, a post-translational modification involving the addition of a carbohydrate to a protein or lipid, and its dysregulation have gained recognition as major players in cancer and metastasis more recently than some other metabolic derangements. Aerobic glycolysis, and in turn increased pools of fructose 6-phosphate, has the potential to fuel hexosamine biosynthesis. The final product, UDP-GlcNAc, serves as the base for elaboration of both N- and O-glycans; however, limited experiments have suggested that it is tightly regulated [14,78]. Membrane-bound as well as secreted proteins may be glycosylated with a wide variety of carbohydrate conformations, and the inherent heterogeneity of the resultant products has slowed their understanding compared to the templated processes involved in protein production. Significant alterations in surface glycan conformation, increased fucosylation or sialyation, as well as changes in glycan secretion, have been identified in numerous cancers including NB, with connections to *MYCN* beginning to be identified [11,15,79,80]. For example, Bley et al. recently analyzed mRNA sequencing results from *MYCN*-amp and non-amp NBs and identified over 30 differentially expressed glycosyltransferase genes between the two [81]. Moreover, Zhu et al. showed increased core fucosylation within neuroblastic regions of *MYCN*-amp tumors using matrix-assisted laser desorption ionization mass spectrometry imaging (MALDI-MSI) and subsequently identified increased expression of a key enzyme responsible for de novo GDP-fucose synthesis as a result of *MYCN* amplification [11].

N-glycosylation, the addition of an oligosaccharide to an asparagine residue, is a complex process beginning with a dolichol-phosphate oligosaccharide precursor and dependent on a large family of coordinated enzymes in the endoplasmic reticulum and Golgi [14]. A study comparing only two NB cell lines, one *MYCN*-amplified and one not, found that *MYCN*-amplified cells tended to display larger glycans, with a preference for sialic acid modifications [82]. Interestingly, a study of ganglioneuroblastoma and NB patients compared with matched controls found increased branched, sialylated N-linked glycans in the serum of NB patients, suggesting utility for serum glycan profiling; however, the study did not stratify by *MYCN* status [80].

O-glycosylation, which involves adding an oligosaccharide to a serine or threonine residue, is crucial for forming H antigen precursor of blood group-related antigens, some of which participate in selectin binding and, ultimately, tumor cell extravasation and metastasis [14,83]. Compared to some adult tumors, little is known about how aberrant O-glycosylation might affect NB progression or prognosis. In vitro, expression of Lewis glycan family members, and glycosyl and fucosyltransferases involved in their construction, tends to be higher in *MYCN*-amp cell lines than in non-amp lines [83], suggesting a role for these structures in more aggressive tumor presentations. However, a prior report using immunohistochemistry to identify blood group-related antigens on tumors of epithelial, neuroectoderm, and mesodermal origin found no expression of these antigens when analyzing a small number on frozen NBs and very limited work has been done since [84].

Core β1,3 galactosyltransferase (C1GALT1) catalyzes formation of the core 1 structure in GalNAc-type O-glycosylation, necessary for more complex core 2 and extended structures [85]. Its expression is increased in numerous adult cancers and correlates with poor outcomes therein. However, C1GALT1 is positively associated with survival in NB cases, even those with *MYCN* amplification, and loss of this enzyme promotes malignant behaviors in NB cells in vitro and in vivo [86]. This enzyme participates in GalNAc O-glycosylation of TrkA, a receptor tyrosine kinase necessary for sympathetic nervous differentiation, and TrkA activation suppresses *MYCN* expression [32,86,87]. When compared to the numerous tumors in which C1GALT1 seems to be tumor-promoting, these data suggest a unique role for O-glycosylation in NB that should be further explored.

### 6.1. Fucosylation

L-fucose is found on N- and O-linked glycans glycolipids, and directly conjugated to proteins via O-fucosylation of serine residues. It plays important roles in processes such as cell motility and angiogenesis. Dysregulated fucosylation, a known entity in neoplasms as diverse as glioblastoma, hepatocellular carcinoma, melanoma, and breast cancer, has been implicated in aggressive tumor growth and is used as a biomarker in some cases [88,89]. Fucose may be derived from exogenous sources of fucose, mannose, or glucose, or it may be taken in by recycling existing glycoconjugates. Regardless of source, it undergoes a multi-step reaction to form GDP-fucose prior to glycan addition. The addition of GDP-fucose to glycan structures relies on a set of fucosyltransferases (FUTs) and transporters thought to discriminate the heritage of their individual substrates [90]. While α-1,2/3/4 fucose linkages may be formed by multiple FUTs, core fucosylation, an α-1,6 linkage to the innermost asparagine-linked GlcNAC, is accomplished solely by FUT8 [14].

Relatively little is known about the role of fucosylation in NB. Zhu et al. demonstrated a pathogenic role for *MYCN*-enhanced overexpression of GDP-mannose 4,6 dehydratase (*GMDS*), the first committed enzyme step in de novo GDP-fucose synthesis, and found significantly increased core fucosylation in *MYCN*-amp neuroblastic tumor regions in situ [11]. This suggests a significant role for FUT8 or perhaps the enzymes that precede it in GDP-fucose biosynthesis, such as GMDS, in NBs. Indeed, FUT8 expression is negatively associated with overall survival in NB; however, this enzyme was not associated with disease staging or *MYCN* status [11]. Taken together, these findings highlight core fucosylation as a critical but understudied modification in NB, with FUT8 and upstream enzymes such as GMDS emerging as potential drivers of tumor aggressiveness. Defining the cell surface and secreted proteins that depend on this modification for their oncogenic potential represents a rational next step in advancing this area of investigation.

### 6.2. Sialylation

Sialic acids are highly abundant cell surface sugars that provide a negatively charged cap on glycans. A wide variety of structures exists, with the most common form found in humans designated Neu5Ac [14]. Oligosialic acids, up to three residues, are commonly found on gangliosides, while polysialic acid, a polymer of sometimes more than 100 residues in a chain formation, is less common but may be found on some structures important to NB, including neural cell adhesion molecule (NCAM) [14]. Polysialylated NCAM may facilitate NB cell migration [91], and serum levels have been suggested to indicate undifferentiated or high-stage disease [92]; however, a study using paraffin-embedded NB samples found no correlation of polysialylated NCAM with *MYCN* status [93].

One area of NB research and treatment in which sialic acid has gained significant favor is the disialoganglioside GD2, which is at present the only approved immunotherapy target for NB patients (dinutuximab). The structure, synthesis, and biology of GD2 are well reviewed in reference [94]. Compared to normal tissues, NBs express high levels of gangliosides, which are compound lipids that contain a carbohydrate moiety, an alcohol called sphingosine, a fatty acid, and at least one sialic acid [95]. In general, ganglioside expression in nervous tissues progresses from simple types such as GD2 toward more complex species from embryonic development through maturity [96]. This pattern becomes dysregulated in *MYCN*-amp NB, where the oncogene drives an adrenergic regulatory program, thus promoting persistence of the immature neuroblast state and high GD2 levels [33,34,97,98]. NBs may be very plastic and able to reversibly transition from one state to another in response to environmental factors [99,100]. In addition to being linked to chemotherapy resistance, this plasticity has been reported to influence GD2 surface display in commonly used cell lines, with a more mesenchymal state leading to lower GD2 expression by throttling GD3 synthase (*ST8SIA1*) expression [101,102].

GD2 has multiple functions in NB, including adhesion to extracellular matrix proteins and providing an anti-phagocytic signal by binding phagocyte Siglec 7 [103,104]. It is also shed by undifferentiated NB cells, and plasma GD2 concentrations are thought to be indicative of active disease [105]. A recent study proposing use of circulating GD2 as a biomarker found high serum concentrations in children with high-risk, *MYCN*-amp, or high-stage disease, suggesting that monitoring circulating GD2 could have utility as a prognostic indicator and treatment response [79]. GD2 expression on tumors may be somewhat heterogeneous, with infrequent tumors initially presenting as partially or fully GD2 negative. Loss as a result of treatment is relatively rare but should be considered in cases of relapse with previous anti-GD2 immunotherapy [106,107,108]. Taken together, the aforementioned studies indicate clinical utility for individual ganglioside profiling alongside *MYCN* status prior to initiating anti-GD2 therapy.

### 6.3. Modulation of Carbohydrate Display

The utility and relatively straightforward nature of carbohydrate modification in cancer treatment is only beginning to be realized. Fucose and sialic acid biosynthesis, as well as some fucosyl and sialyltransferases, can be inhibited by treatment with unnatural monosaccharide derivatives including Fucotrim I, 2-fluorofucose, FNANA, P-SiaFNEtoc, and others [109,110]. Zhu et al. showed that oral 2-fluorofucose treatment of immune-deficient mice after tumor initiation with *MYCN*-amp BE(2)-C cells significantly slowed tumor growth, thus indicating that fucosylation blockade may have considerable benefit in these aggressive tumors [11]. More work remains to be done in this area, however, including significant study into how the tumor’s immune microenvironment is altered with reduced fucosylation. Exogenous methods to enhance *ST8SIA1* and GD2 expression are also an active area of investigation. The combination of peracetylated sialic acid and vorinostat, an HDAC inhibitor, effectively boosted GD2 expression in *MYCN*-amp cells, suggesting utility for this treatment combination for in vivo investigations [111]. Inhibition of *MYCN*- driven histone methyltransferase EZH2 has also been shown to upregulate GD2 in a mouse mesenchymal NB xenograft and, moreover, EZH2 and HDAC inhibition were synergistic in vitro, promoting expression of tumor suppressor genes in *MYCN*-amp cells, thus demonstrating that multiple routes to therapeutic benefit exist in this realm [102,112].

## 7. Conclusions

There is now abundant evidence from both in vitro models and in vivo preclinical systems that *MYCN* serves as a master regulator of metabolic transformation in NB. Its control over glycolysis, glutaminolysis, and fatty acid metabolism positions it as a central orchestrator of the biosynthetic and energetic programs that sustain tumor growth [113]. Traditional in vitro models, while indispensable, are often limited by supraphysiologic glucose and glutamine concentrations and non-physiologic oxygen content, conditions that may distort metabolic dependencies relative to the native tumor microenvironment [114]. However, the expanding toolkit of both spatial and stable isotope-resolved metabolomics (SIRM) now enables these regulatory networks to be mapped within the authentic context of human NB tissue and patient-derived samples, offering an opportunity to connect molecular rewiring, genetic features, and clinical behavior [16,17]. The capacity to measure direct metabolite flux within pediatric cancers was recently reported by Johnston et al. [39]. In addition, our group recently leveraged MALDI-MSI to profile the N-linked glycome of human NBs [11]. The spatial-preserving nature of this analysis noted marked N-linked glycan heterogeneity throughout the human tumor samples, emphasizing limitations to traditional metabolomic analysis of bulk homogenates. Together, these advances underscore the critical importance of studying neuroblastoma metabolism within the intact human tumor context, where spatial heterogeneity and physiologic conditions can be faithfully captured to reveal clinically actionable vulnerabilities.

Emerging evidence increasingly suggests that *MYCN*’s oncogenic consequences extend beyond core canonical metabolism into glycosylation, a complex post-translational signaling axis. Glycan remodeling represents a critical interface between metabolic rewiring and cell–cell communication, with profound implications for immune recognition, invasion, and therapy resistance. The hexosamine biosynthetic pathway (HBP) exemplifies this integration, serving as a central metabolic shunt supplied by glycolytic, glutamine, and fatty acid-derived intermediates to generate UDP-GlcNAc for N- and O-linked glycosylation. Further details of this integration of the HBP as a central hub is provided in a separate review by Akella et al. [115]. Despite differential end-product abundance manifest by cell surface glycan expression and upstream substrate sources that are *MYCN*-dependent, this dimension of *MYCN* biology remains understudied. Given the role of glycosylation in facilitating cell surface and paracrine interactions, our group views glycosylation as an integrated output of *MYCN*-driven transformation that warrants systematic exploration. Together, these converging areas of investigation position *MYCN* at the nexus of metabolic and post-translational control, reinforcing its role as a central driver of NB biology and a compelling therapeutic target.

Importantly, metabolic and glycosylation-dependent vulnerabilities already intersect with therapies that have transformed the clinical management of high-risk NB. Aberrant sialylation is exploited in anti-GD2–based immunotherapy, an FDA-approved antibody strategy that improves survival in children with high-risk disease. Similarly, the polyamine axis is another *MYCN*-coupled metabolic output that is being leveraged in the clinic using the irreversible ODC1 inhibitor eflornithine (DFMO) to reduce relapse risk following anti-GD2 therapy. These precedents validate the principle that *MYCN*-programmed metabolism is both mechanistically central and therapeutically actionable. A summary of pathway-specific therapeutic interventions is included in Table 2.

NB metabolism is dynamic and heterogeneous. Systematic implementation of approaches such as spatial metabolomics and SIRM to map nutrient flux, quantify pathway end-products, and define regional heterogeneity in situ will enable investigators to move beyond reliance on expression-based surrogates to capture the functional metabolic state of high-risk NBs. These advanced metabolomic strategies may also define the clinical contexts in which known metabolic aberrancies, such as sialylation and polyamine biosynthesis, might be most effectively leveraged, while uncovering new metabolic nodes of oncogenic potential within this devastating and metabolically driven childhood cancer.

## Figures and Tables

**Figure 1 cancers-17-03256-f001:**
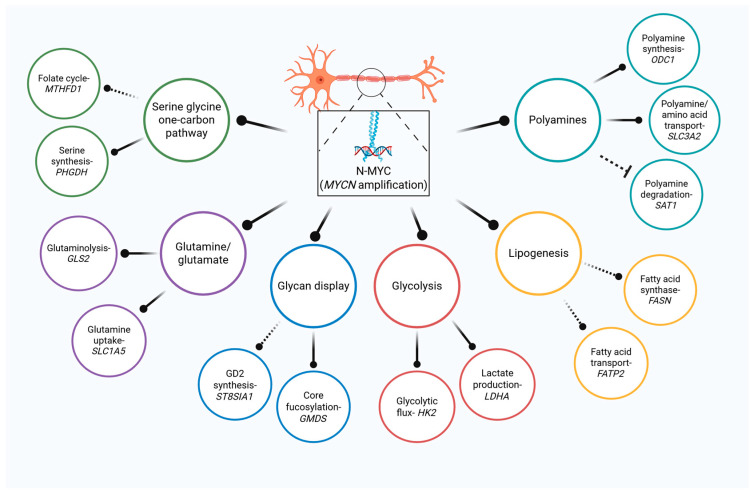
Multi-nodal control of metabolism by N-MYC in NB. MYCN is a master regulator of central metabolism that drives glucose, amino acid, fatty acid, and polyamine metabolism. Complete nodes lines represent direct regulation while indirect regulation is illustrated by dashed connectors.

**Table 1 cancers-17-03256-t001:** Summary of MYCN-influenced metabolic mediators in NB.

Gene (Protein)	Metabolic Node	Function	*MYCN* Effect	Notes	Reference
*HK2* (Hexokinase 2)	Glycolysis (entry)	Converts glucose → glucose-6-phosphate	Transcriptional upregulation; HIF-1α cooperation	Increases glycolytic flux; sensitizes to 2-DG, 3-BP; poor prognosis marker	[18,19]
*LDHA* (Lactate Dehydrogenase A)	Glycolysis (exit)	Converts pyruvate → lactate, regenerates NAD⁺	Transcriptional upregulation; HIF1α cooperation	Marker of poor prognosis; inhibition efficacy context-dependent	[20,21]
*GLS2* (Glutaminase 2)	Glutaminolysis	Converts glutamine → glutamate	Direct transcriptional activation	Correlates with poor prognosis	[22]
*SLC1A5* (ASCT2)	Glutamine uptake	Imports glutamine into cells	Direct transcriptional activation	Supports glutamine addiction in *MYCN*-amp NB; inhibition reduces growth & sensitizes to GLS blockade; N-Myc coordinates with ATF4	[23]
*PHGDH* (Phosphoglycerate dehydrogenase)	Serine synthesis (SGOC)	Converts 3-PG → 3-phosphohydroxypyruvate	Direct transcriptional activation	Inhibition selectively toxic to *MYCN*-amp NB cell lines	[24]
*MTHFD1* (Methylenetetrahydrofolate dehydrogenase 1)	Folate cycle (one-carbon metabolism)	Provides 1C units for nucleotide/methylation	Increase	Amp cells have elevated folate need. Knockdown triggers apoptosis/redox stress; potential therapeutic target	[25,26]
*ODC1* (Ornithine decarboxylase 1)	Polyamine synthesis	Converts ornithine → putrescine (rate-limiting)	Direct target; often co-amplified with *MYCN*	FDA-approved target (DFMO); high ODC1 = poor prognosis	[27]
*SLC3A2* (CD98hc)	Amino acid & polyamine transport	Neutral AA, cystine, polyamine uptake	Direct *MYCN* target	Supports mTORC1 activation, glutathione synthesis, polyamine salvage; poor prognosis marker	[28]
*SAT1* (Spermidine/spermine acetyltransferase 1)	Polyamine degradation	Acetylates for export/degradation	Indirect *MYCN* effect; suppressed when *MYCN* drives synthesis	Through transcription factor Sp1	[28]
*FASN* (Fatty acid synthase)	Lipogenesis	De novo palmitate synthesis	Indirectly upregulated via *MYCN* → SREBP1/MondoA	Inhibition impairs growth; bypass possible via exogenous FA uptake	[29,30]
*SLC7A2* (FATP2)	Lipogenesis	Fatty acid transport	Increase	Fatty acid uptake in NB; therapeutic vulnerability	[31]
*GMDS* (GDP-mannose 4,6-dehydratase)	Fucosylation	De novo GDP-fucose synthesis	*MYCN* upregulates GMDS	Core fucosylation increased in *MYCN*-amp NB	[11]
*C1GALT1* (Glycoprotein-N-acetylgalactosamine 3-beta-galactosyltransferase 1)	O-glycosylation	Builds core 1 O-glycan (T antigen)		Glycosylates TrkA, indirectly suppressing *MYCN*/promotes differentiation	[32]
*ST8SIA1* (GD3 synthase)	Ganglioside (GD2) synthesis	Catalyzes GD3 synthesis → precursor of GD2	Maintained by *MYCN*-driven adrenergic program	*MYCN* promotes immature adrenergic program/simple gangliosides; High GD2 = target of dinutuximab; expression drops in mesenchymal state	[33,34]

**Table 2 cancers-17-03256-t002:** Pathway-specific metabolic vulnerabilities and therapeutic interventions in neuroblastoma.

Pathway	Key Target(s)	Inhibitor/Drug	Clinical Status	Notes	Reference
Polyamine metabolism	ODC1 (Ornithine decarboxylase)	DFMO (eflornithine/Iwilfin)	FDA-approved for high-risk NB maintenance post anti-GD2 therapy	First metabolic therapy approved in NB; reduces relapse risk; combinations under study (e.g., DFMO + celecoxib [SAT1 induction], DFMO + AMXT-1501 [polyamine transport blockade])	[27,28,59,60,61,64,65,66,67,68]
	Polyamine transporters (SLC3A2, ATP13A3)	AMXT-1501	Phase I/II trials (adult cancers); preclinical NB	Enhances DFMO efficacy by blocking salvage pathway	[62,63]
Glycolysis	HK2	2-Deoxyglucose (2-DG), 3-bromopyruvate (3-BP)	Preclinical	*MYCN*-amp NB cells sensitive to 2-DG; synergy with mTOR inhibitors	[18,19]
	LDHA	FX11, Oxamate	Preclinical	Inhibition reduces lactate flux and proliferation; context-dependent efficacy due to metabolic flexibility	[21,44,45]
Glutamine metabolism	ASCT2 (SLC1A5), GLS2	V-9302 (ASCT2 inhibitor), CB-839 (telaglenastat, GLS inhibitor)	Preclinical NB; CB-839 in clinical trials for adult cancers	NB cells show glutamine addiction; inhibition induces apoptosis in *MYCN*-amp models	[53]
Serine/One-Carbon (SGOC) metabolism	PHGDH, MTHFD1/2, SHMT2	PHGDH inhibitors (NCT-503)	Preclinical NB	*MYCN* upregulates SGOC genes; targeting disrupts nucleotide synthesis, redox balance, epigenetics	[24]
Lipid metabolism	FASN (Fatty acid synthase)	Orlistat, UB006	Preclinical	FASN inhibition reduces xenograft growth; bypass via exogenous fatty acid uptake	[29]
Glycosylation	GMDS/FUT8 (core fucosylation)	2-fluorofucose (2FF)	Preclinical NB	Inhibits *MYCN*-amp xenograft growth	[11]
	ST8SIA1 (GD3 synthase → GD2)	Dinutuximab (anti-GD2 mAb)	FDA-approved for high-risk NB	Anti-GD2 immunotherapy improves survival; cornerstone of NB treatment	[102,103]
Epigenetic–metabolic crosstalk	EZH2, HDACs (regulate GD2 expression)	Tazemetostat (EZH2 inhibitor), Vorinostat (HDAC inhibitor)	Clinical trials in NBs	Can enhance GD2 expression; synergize with anti-GD2 immunotherapy	[102,111,112]

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
