# Peer review of "MYCN-Driven Metabolic Networks Are a Critical Dependency of High-Risk Neuroblastomas"

_cancers, 2025, doi:10.3390/cancers17193256_

Round 1
Reviewer 1 Report
Comments and Suggestions for Authors
The manuscript is well written, comprehensive, and logically structured. It is written with clarity and provides valuable insights into how MYCN rewires metabolism to sustain tumor growth and survival. The authors summarize the published literatures and highlight potential novel avenues for therapeutic intervention. This work is a useful reference for researchers in the field.
Author Response
No comments
Reviewer 2 Report
Comments and Suggestions for Authors
This is an excellent and timely review that comprehensively summarizes current knowledge on MYCN-driven metabolic reprogramming in neuroblastoma, with a particular emphasis on the integration of canonical metabolic pathways, glycosylation, and translational opportunities. The manuscript is well written, logically structured, and provides both mechanistic insight and clinical perspective. I believe it will be of high interest to the readership.
I have no major concerns. If the authors wish to further improve the clarity of the text, I would only suggest a few very minor points:
Some sentences in the conclusions are rather dense; breaking them into shorter segments may improve readability.
In Tables 1 and 2, the formatting is somewhat misleading. A horizontal enlargement of the table should allow the longest information to fit into a maximum of 2-3 lines, making each line more readable.
A few minor typographical issues (e.g., line 434: “but and therapeutically actionable” should probably read “both mechanistically central and therapeutically actionable”).
These are optional refinements and do not affect my overall positive assessment.
Author Response
Comment 1. Some sentences in the conclusions are rather dense; breaking them into shorter segments may improve readability.
Response- Agreed, and thank you for pointing this out. Some sentences in the introduction and conclusions have been altered for the purposes of clarity, with no changes to substance.
Comment 2. In Tables 1 and 2, the formatting is somewhat misleading. A horizontal enlargement of the table should allow the longest information to fit into a maximum of 2-3 lines, making each line more readable.
Response- You are right. We will resubmit these tables with a landscape layout.
Comment 3. A few minor typographical issues (e.g., line 434: “but andtherapeutically actionable” should probably read “both mechanistically central and therapeutically actionable”)
Response- Fixed- thank you.
Reviewer 3 Report
Comments and Suggestions for Authors
The authors provide an interesting summary of the affect of MYCN amplification on protein and carbohydrate metabolism. For investigators who may not remember all of the metabolic pathways discussed, a schematic or two indicating catabolic and anabolic steps involved that are affected by MYCN would be helpful and should be included.
More specifically:
- Table 1: Indicate whether expression of each MYCN-amp affected enzyme or transporter was identified in human NB tumor samples or if cultured NB cells indicate cell line.
- Line 98: Include a reference(s) for the more recent work referred to.
- Line 170: Does elevation of the proteins by activation of the SGOC pathway in MYCN-amp cases correlate with treatability of patients with high-risk NB in the absence of MYCN-amp?
- Line 193: Instead of just saying few therapies targeting amino acid dependencies have been developed, include a brief discussion of what problems might exist between studying them in cultured cells and using them clinically. Also, in the examples given, describe NB cells used in the studies.
- Lines 212-213: Statement is made that DFMO therapy increases both event-free and overall survival in high-risk cases, indicate, if known, whether that affect is for both high risk NBs expressing MYCN-amp and those that don’t.
- Line 260: At the end of the paragraph indicate potential approaches/problems for expanding the pharmacological toolkit. Could combine this with the comment made for line 193.
- Line 316: Believe the word “in” needs to be inserted between implicated and aggression
- Line 434: As written the word “but” needs to be deleted.
- While MYCN-amp has the affects discussed, would any of the possible treatment points apply to patients with high risk NB minus MYCN-amp?
- Since the authors indicate, lines 76-78, that this review is for both clinicians and researchers, reference to NB data bases per se should be included.
Author Response
Comment 1. Table 1: Indicate whether expression of each MYCN-amp affected enzyme or transporter was identified in human NB tumor samples or if cultured NB cells indicate cell line.
Response: We appreciate that this would be helpful to have in the table; however, the table is already rather crowded. Most of this information is available in the text and in a few cases, we have added an indication of provenance at your suggestion.
Comment 2. Line 98: Include a reference(s) for the more recent work referred to.
Response: This line has been referenced and combined with NB-specific work that was published while this document was in review. Both were moved to the vicinity of line 142 for the sake of clarity.
Comment 3. Line 170: Does elevation of the proteins by activation of the SGOC pathway in MYCN-amp cases correlate with treatability of patients with high-risk NB in the absence of MYCN-amp?
Response: Reference 39 (Xia, 2019) attempted to look at this question with analysis of primary tumor expression data. They found that the SGOC gene signature, which includes PHGDH, was slightly elevated in high-risk non-amp tumors, while it was very significantly elevated in high-risk MYCN-amp tumors. They further showed that higher expression of the SGOC genes is associated with advanced stage tumors and lower survival overall; however, the strongest association appears to be with MYCN-amplification. In vitro and in vivo trials demonstrate increased susceptibility to NCT-503, a small molecule inhibitor of PHGDH, within the MYCN-amplified cell lines, suggesting that MYCN-amplified NBs are preferentially sensitive the SGOC targeted approaches. We have updated the text to highlight these findings.
Comment 4. Line 193: Instead of just saying few therapies targeting amino acid dependencies have been developed, include a brief discussion of what problems might exist between studying them in cultured cells and using them clinically. Also, in the examples given, describe NB cells used in the studies. Line 260: At the end of the paragraph indicate potential approaches/problems for expanding the pharmacological toolkit. Could combine this with the comment made for line 193.
Response: Line 193: We agree with this comment and have added the following commentary “A major challenge to the translational integration of targeting amino acid dependencies is that much of this foundational work derives from studies in two-dimensional cultured cell systems, where nutrient availability, oxygen tension, and immune interactions differ dramatically from the tumor microenvironment in vivo and in situ. For instance, the composition of amino acids and redox buffers in standard culture media (e.g., high cystine, glutamine, and glucose) does not accurately reflect the nutrient gradients, stromal interactions, or hypoxic conditions present within bulky NBs. As a result, metabolic dependencies observed in vitro may be exaggerated, or alternatively masked, compared to what occurs in patients, highlighting the need to develop physiologically relevant NB culture systems and highlighting the importance of validating cell culture within disease relevant models in vivo.
Line 260: We agree with this comment. We have added the following commentary to this section. “Despite this promise, the current pharmacological toolkit remains underdeveloped. Expanding this toolkit will require both adaptation of existing metabolic inhibitors from other cancer settings and development of NB-tailored approaches guided by in vivo validation. However, major challenges remain, including limited specificity, systemic toxicity given the importance of lipid metabolism in normal tissues, and the intrinsic redundancy of metabolic networks that allows NB cells to adaptively rewire their fuel utilization and bypass single-pathway inhibition. This metabolic plasticity raises the risk that inhibition of a single pathway may be insufficient to achieve durable responses. Overcoming these barriers will necessitate rational combination strategies, improved drug design, and integration of metabolic biomarkers into translational studies.”
Comment 5. Lines 212-213: Statement is made that DFMO therapy increases both event-free and overall survival in high-risk cases, indicate, if known, whether that affect is for both high risk NBs expressing MYCN-amp and those that don’t.
Response: Sholler et al. 2018 showed that DFMO treatment increased survival for both groups. This has been added to the text.
Comment 6. Line 316: Believe the word “in” needs to be inserted between implicated and aggression.
Response: Fixed- thank you.
Comment 7. Line 434: As written the word “but” needs to be deleted.
Response: Fixed- thank you.
Comment 8. While MYCN-amp has the affects discussed, would any of the possible treatment points apply to patients with high risk NB minus MYCN-amp?
Response: This is an excellent point but in our perspective beyond the scope of the manuscript in its current format. The authors intent here was to take the lens of MYCN-amplification in NB and highlight its central role of metabolism with the perspective of therapeutic benefit. We have attempted to highlight instances where evidence supports therapeutic sensitivity of non-MYCN-amplified NBs throughout the manuscript.
Comment 9. Since the authors indicate, lines 76-78, that this review is for both clinicians and researchers, reference to NB data bases per se should be included.
Response: We appreciate the reviewer’s suggestion regarding inclusion of specific NB databases. However, we feel that citing individual databases is not fully aligned with the scope of our review, for several reasons. First, our intent is to synthesize and contextualize the biological and translational insights relevant to MYCN-driven metabolic vulnerabilities, rather than to provide an exhaustive catalog of informatic resources. The primary audience of this review, including both clinicians and researchers, will benefit more from an integrated conceptual framework of how MYCN rewires tumor metabolism and how this may influence therapeutic strategies, rather than database navigation guidance.
Second, there is no single NB database that captures the breadth of relevant molecular, clinical, and preclinical data. Available repositories (e.g., SEQC, TARGET) differ in format, depth, and accessibility, and require considerable technical expertise to analyze appropriately. Highlighting one or two could inadvertently bias the reader or suggest endorsement of a particular platform, while listing several risks distracting from the main narrative.
Finally, the literature we cite already incorporates results derived from these datasets, ensuring that the findings most relevant to clinical translation are included. In this way, readers are exposed to the biologic conclusions and therapeutic implications without the added complexity of navigating raw database infrastructure